# Enhancement of ^18^F-Fluorodeoxyglucose PET Image Quality by Deep-Learning-Based Image Reconstruction Using Advanced Intelligent Clear-IQ Engine in Semiconductor-Based PET/CT Scanners

**DOI:** 10.3390/diagnostics12102500

**Published:** 2022-10-15

**Authors:** Ken Yamagiwa, Junichi Tsuchiya, Kota Yokoyama, Ryosuke Watanabe, Koichiro Kimura, Mitsuhiro Kishino, Ukihide Tateishi

**Affiliations:** Department of Diagnostic Radiology and Nuclear Medicine, Tokyo Medical and Dental University, 1-5-45 Yushima, Bunkyo-ku, Tokyo 113-8510, Japan

**Keywords:** image quality, deep learning reconstruction, semiconductor-based PET/CT, ^18^F-fluorodeoxyglucose positron emission tomography

## Abstract

Deep learning (DL) image quality improvement has been studied for application to ^18^F-fluorodeoxyglucose positron emission tomography/computed tomography (^18^F-FDG PET/CT). It is unclear, however, whether DL can increase the quality of images obtained with semiconductor-based PET/CT scanners. This study aimed to compare the quality of semiconductor-based PET/CT scanner images obtained by DL-based technology and conventional OSEM image with Gaussian postfilter. For DL-based data processing implementation, we used Advanced Intelligent Clear-IQ Engine (AiCE, Canon Medical Systems, Tochigi, Japan) and for OSEM images, Gaussian postfilter of 3 mm FWHM is used. Thirty patients who underwent semiconductor-based PET/CT scanner imaging between May 6, 2021, and May 19, 2021, were enrolled. We compared AiCE images and OSEM images and scored them for delineation, image noise, and overall image quality. We also measured standardized uptake values (SUVs) in tumors and healthy tissues and compared them between AiCE and OSEM. AiCE images scored significantly higher than OSEM images for delineation, image noise, and overall image quality. The Fleiss kappa value for the interobserver agreement was 0.57. Among the 21 SUV measurements in healthy organs, 11 (52.4%) measurements were significantly different between AiCE and OSEM images. More pathological lesions were detected in AiCE images as compared with OSEM images, with AiCE images showing higher SUVs for pathological lesions than OSEM images. AiCE can improve the quality of images acquired with semiconductor-based PET/CT scanners, including the noise level, contrast, and tumor detection capability.

## 1. Introduction

^18^F-Fluorodeoxyglucose positron emission tomography/computed tomography (^18^F-FDG PET/CT) is widely used for the diagnosis of neoplastic diseases. It helps differentiate between benign and malignant lesions, determine the stage of cancers, and plan treatment methods [1,2,3,4,5,6]. It is used to diagnose ischemic, inflammatory, and degenerative diseases [7,8,9,10,11,12]. However, one of the main disadvantages of ^18^F-FDG PET/CT is its low resolution. Some methods proposed to obtain high-resolution PET images include increasing the acquisition time, using a time-of-flight technique, new reconstruction methods, and semiconductor-based PET/CT scanners [13,14,15]. In addition, silicon photomultiplier (SiPM)-based detectors (in semiconductor-based PET/CT scanners) have several advantages compared with photomultiplier tubes, such as a smaller size, higher intrinsic time resolution, and higher photon detection efficiency [15,16,17,18]. Van Sluis et al. reported that on semiconductor PET/CT images, the lesion demarcation was sharper, the overall image quality was higher, and the visually assessed signal-to-noise ratio was higher than on conventional PET/CT images [19].

In recent years, convolutional neural networks have been applied to different medical imaging technologies, including CT, magnetic resonance imaging, and PET/CT [20,21,22,23,24]. Advanced Intelligent Clear-IQ Engine (AiCE, Canon Medical Systems, Tochigi, Japan) is a commercialized deep-learning-based reconstruction (DLR) tool for PET/CT imaging. In our previous study, we found that the DLR method improves image quality compared with conventional imaging using Gaussian filters, by providing a clearer definition of tumor lesions, higher overall image quality, and higher visually assessed signal-to-noise ratio. Regarding semi-quantitative image quality, we found that the standardized uptake value (SUV) on images using the DLR method was higher in tumors and healthy tissues in small organs [25].

To the best of our knowledge, no previous study has evaluated differences in image quality between standard semiconductor PET/CT imaging and the corresponding images obtained with application of the DLR method. The main purpose of this study was to compare both visually and semi-quantitatively the clinical images obtained using the DLR method and the standard semiconductor PET/CT images.

## 2. Materials and Methods

### 2.1. Patients

The study population consisted of all patients who underwent ^18^F-FDG PET/CT between 6 May 2021 and 19 May 2021 at the Department of Nuclear Medicine and Molecular Imaging at Tokyo Medical and Dental University. Patients were excluded if they had a glucose level greater than 200 mg/dL or could not lie still during the scan. The Institutional Review Board of Tokyo Medical and Dental University approved this study, and written informed consent was obtained from each patient.

### 2.2. PET/CT Imaging

All patients fasted for at least 6 h before intravenous administration of ^18^F-FDG, and glucose levels were evaluated. The maximum blood glucose level observed was 165 mg/dL. The image acquisition was performed approximately 60 min after the injection of 3.7 MBq/kg ^18^F-FDG. A PET/CT scanner (Cartesion Prime, Canon Medical Systems, Tochigi, Japan) was used to scan the patients from the crown of the head to the mid-thigh. The CT parameters used for attenuation correction were as follows: tube voltage, 120 kV; field of view, 700 mm; pitch factor, 0.813; helical pitch, 65; and slice thickness, 2 mm. Following CT image acquisition, PET images were acquired in 90 s per bed, and the matrix size was 336 × 336. The DLR images were reconstructed using AiCE (Canon Medical Systems). AiCE network consists of an 8-layer deep convolutional neural network as shown in Figure 1, and it is designed to yield a high-quality image such as very long scan duration images when regular scan duration data are input. To train the network, the listmode data of long duration were scanned. Then, the images reconstructed full duration of listmode data are used as training targets and images obtained by reconstructing time-split listmode are used as training input. The OSEM PET images with 3 mm Gaussian filters were reconstructed using two iterations and 12 subsets with a point-spread function. These protocols were the same as those used in clinical practice and were determined based on a phantom test in accordance with FDG-PET/CT procedure guidelines in Japan [26].

### 2.3. Qualitative Analysis

The acquired images were independently reviewed and analyzed using Vox-base SP1000 workstation (J-MAC Systems, Sapporo, Japan). Two experienced nuclear medicine physicians (with 16 and 9 years of experience interpreting PET scans, respectively) blindly evaluated all PET images for qualitative analysis.

The image quality rating was based on 5-point scales using the following quality criteria: tumor delineation, ranging from 1 (pathological lesion cannot be confirmed) to 5 (excellent lesion margin delineation); overall image quality, ranging from 1 (poor overall image quality) to 5 (excellent overall image quality); and image noise, ranging from 1 (largely interfering noise) to 5 (no relevant noise perceivable). In cases of large rating differences between readers, consensus was obtained through meetings.

### 2.4. Quantitative Analysis

For the semi-quantitative analyses, 0.5-mL spherical volumes of interest (VOIs) were placed in healthy organs, including the parotid glands, lungs, aortic arch, left ventricle, liver, spleen, and quadriceps muscles. Standardized uptake value (SUV) is the ratio of radioactivity in a VOI to injected dose per patient’s body weight. It is calculated as:$$SUV=\frac{PET\;count\left(cps\right)\times Cross\;calibration\;factor\left(Bq/cps/ml\right)}{\frac{Injected\;dose\left(Bq\right)}{Body\;weight\left(g\right)}}$$

Three SUV parameters (SUVmax, SUVmean, and SUVpeak) were obtained from these VOIs. SUVmean is the average SUV and SUVmax is pixel with largest SUV SUVpeak is calculated by defining a 1 cm^3^ spherical volume at every pixel within the selected VOI, measuring the average SUV in each sphere and then calculating the maximum of all the average SUVs across all these spheres [27]. We also placed VOIs in up to five pathological lesions per patient and measured the same three parameters. For calculating the noise level, we placed 30 mm diameter VOI at the right liver lobe and determined the liver signal-to-noise ratio (SNR) as the SUVmean divided by the standard deviation.

In addition, we counted the number of pathological lesions with increased uptake of radiotracers in AiCE and OSEM images to evaluate detection capability.

### 2.5. Statistical Analysis

The scores for qualitative analysis were compared between the two reconstruction methods using a two-tailed paired-sample t-test. For the inter-reader agreement, we reassigned the original 5-point scores to 3-point scores (1 and 2 reassigned as 1, 3 as 2, and 4 and 5 as 3). The SUV parameters for healthy organs and pathological lesions were compared between the two reconstruction methods using a two-tailed paired-sample t-test. All statistical analyses were performed using SPSS Statistics version 24 (IBM, Armonk, NY, USA). Statistical significance was set at *p* < 0.05.

## 3. Results

The study population included 30 patients with a mean age of 65.3 years (range 25–84). Table 1 summarizes the patient demographics and clinical data. Three patients had two pathologies; therefore, the total number of diseases was 33.

In the qualitative analysis, the images reconstructed using AiCE obtained significantly higher scores than the OSEM images in terms of tumor delineation, image noise, and overall image quality (Table 2). The overall inter-reader agreement showed a Fleiss kappa value of 0.57.

Table 3 shows the SUV parameters of healthy tissues. Among 21 parameters, 11 were significantly different between AiCE and OSEM images. Of the 11 parameters, 10 were higher for OSEM images than for AiCE images, and 1 parameter was higher for AICE images than for OSEM images. We detected pathological lesions in the pituitary gland, tonsils, parotid glands, lungs, breasts, thyroid gland, liver, gallbladder, pancreas, adrenal glands, uterus, prostate, lymph nodes (cervical, mediastinal, supraclavicular, para-aortic, mesentery, and pelvic), aorta, bones (humerus, vertebrae, ribs, ilium, sacrum), anterior mediastinum, frontal sinuses, cervical spinal cord, dural canal, abdominal cavity, and subcutaneous areas. All SUV parameters were higher when measured for pathological lesions on AiCE images than when measured for them on OSEM images, and the difference was statistically significant for SUVmax and SUVpeak (Table 4). The liver SNRs in AICE images were significantly higher than that of OSEM images (OSEM: 10.82 ± 2.12 vs. AiCE: 15.98 ± 3.15; *p* < 0.05).

Table 5 shows the results of the comparison of detection capability. All 30 patients had pathological lesions which were detected using AiCE images. One patient had no lesions detected in conventional images. Among the 29 patients with lesions detected on OSEM images, 6 had more lesions detected on AiCE images. Thus, in seven patients (23.3%), more lesions were identified using AiCE images, with a significant difference in the number of lesions detected by each reconstruction method (OSEM: 4.27 ± 6.42 vs. AiCE: 4.60 ± 6.68; *p* = 0.03). Among these seven patients, two had bone lesions, two lymph node lesions, one had a skin nodule, one had an adrenal nodule, and one had a lung lesion. Representative cases are shown in Figure 2 and Figure 3. Figure 3 shows the lesions detected on the AiCE images and missed on the OSEM images.

## 4. Discussion

In this study, we investigated improvements in image quality using AiCE (DLR method) for semiconductor FDG-PET/CT imaging.

In the qualitative assessment, the images obtained using the AiCE had a significantly higher quality than the OSEM images. Van Sluis et al. found that semiconductor PET images were superior to conventional PET images in terms of lesion demarcation, visually assessed signal-to-noise ratio, and overall image quality [19]. Here, we demonstrate that AiCE can further improve the quality of semiconductor PET images.

In healthy organs, 11 of 21 SUV parameters (52.4%) showed significant differences between AiCE and OSEM images. These differences were mainly due to the higher SUVmax and SUVpeak of OSEM images in larger organs, such as the liver, spleen, and quadriceps muscles. This result is probably due to edges being denoised, as observed in our previous study [25]. In large organs, both AiCE and OSEM methods remove the noise without smoothing out the structure. Since AiCE can denoise slightly more than OSEM, SUVmax and SUVpeak are smaller for AiCE while having the same SUVmean. However, in smaller organs, the OSEM method smears out the structure of the organs. This results in a smaller SUVmean, SUVmax, and SUVpeak for OSEM.

In pathological lesions, AiCE images showed higher SUVmax and SUVpeak values. This result is consistent with the findings of our previous study. Some other studies suggest that semiconductor PET reduces the partial volume effect and increases SUVmax, especially in small lesions [28,29]. Economou et al. speculate that the higher SUVmax noted in small lesions on SiPM-based PET/CT images is most likely due to a new reconstruction algorithm [30]. In contrast, our semiconductor-based PET/CT scanner, Cartesion Prime, uses a Gaussian filter rather than the new reconstruction algorithm. This method may have blurred the distinction between two closely adjacent objects on the OSEM images. We also demonstrated that AiCE images were denoised better than OSEM images based on their higher liver SNR.

Among the 30 patients considered, AiCE images detected more pathological lesions in 7 patients. These lesions were in various organs, including the liver and lungs. In one of the seven patients, we found bone metastasis (on the sacrum) that was not detected on OSEM images. This result may be due to the increase in the SUVmax and SUVpeak values of the lesions on AiCE images without a significant increase in the SUV parameters of most healthy organs.

## 5. Conclusions

Our qualitative analysis showed that AiCE (DLR method) is superior to the OSEM method in terms of lesion delineation, overall image quality, and image noise. In terms of semiquantitative image quality, several SUV parameters in healthy organs were reduced on AiCE images because of denoising. In contrast, the AiCE method increased the SUV parameters in pathological lesions, whereas Gaussian filtering decreases the SUV by blurring. As for lesion detection capabilities, AiCE detected more pathological lesions, and none were detected only on OSEM images. Our results demonstrate that the AiCE (DLR method) significantly reduces noise compared with Gaussian filtering, without losing the quantitative information of PET images, such as SUVmax and SUVmean. Future studies that include a more homogeneous group of patients are needed to evaluate the clinical utility of AiCE on PET images.

## Figures and Tables

**Figure 1 diagnostics-12-02500-f001:**
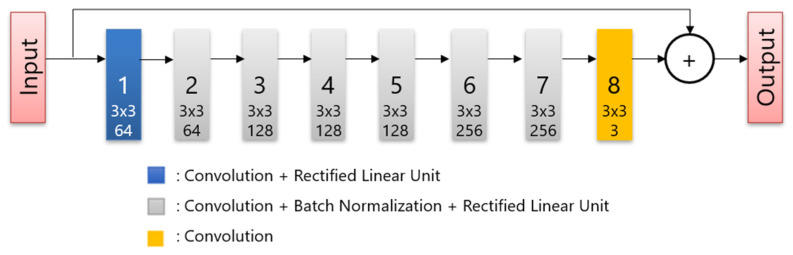
The structure of the deep convolutional network.

**Figure 2 diagnostics-12-02500-f002:**
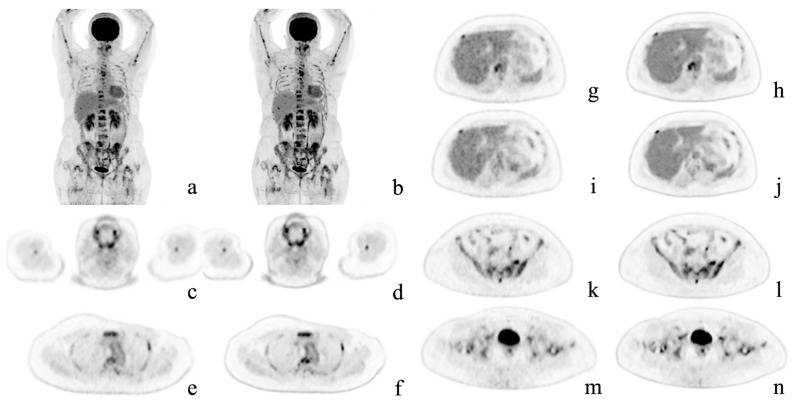
Representative case. A 54-year-old woman with multiple bone metastases to the spine, ribs, sternum, humerus, hipbone, and femur after surgery for left breast cancer. Maximum intensity projection and axial positron emission tomography images with conventional Gaussian filtering (**a**,**c**,**e**,**g**,**i**,**k**,**m**) and those obtained with deep learning reconstruction (**b**,**d**,**f**,**h**,**j**,**l**,**n**). AiCE (DLR) images show a reduction in noise, especially in the liver, and are easier to identify the metastases.

**Figure 3 diagnostics-12-02500-f003:**
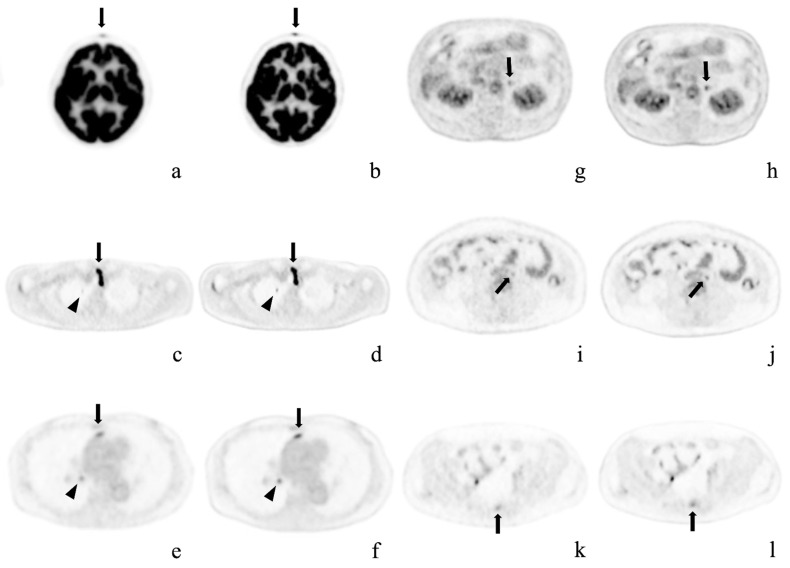
Representative cases. Axial positron emission tomography images with conventional Gaussian filtering (**a**,**c**,**e**,**g**,**i**,**k**) and those obtained with deep learning reconstruction (**b**,**d**,**f**,**h**,**j**,**l**). These pathological lesions were missed or were considered physiological uptake by OSEM imaging alone, but were detected on AiCE (DLR) imaging. (**a**,**b**) Forehead cutaneous nodule (arrows) in a 64-year-old woman; (**c**,**d**) Right superior pulmonary lobe (arrowheads) and superior mediastinal lymph node (arrows) in a 58-year-old man after surgery for thyroid cancer; (**e**,**f**) Anterior mediastinal (arrows) and right hilar (arrowheads) lymph node metastases in a 73-year-old woman with right inferior lung cancer; (**g**,**h**) Left adrenal adenoma in an 80-year-old man; (**i**,**j**) Enlarged para-aortic lymph node in a 41-year-old woman; k, l) Sacral metastasis (arrows) in the same patient.

**Table 1 diagnostics-12-02500-t001:** Patient demographic and clinical data (*n* = 30).

Age (years)			65.3 ± 13.8
Sex			
	Male		13
	Female		17
Weight (kg)			63.5 ± 16.1
Disease			
	Malignancy		
		Lung cancer	10
		Breast cancer	3
		Pancreatic cancer	3
		Malignant lymphoma	3
		Colon cancer	2
		Tongue cancer	1
		Pharyngeal cancer	1
		Thyroid cancer	1
		Liver cancer	1
		Adrenal cancer	1
		Renal cancer	1
		Bladder cancer	1
		Prostate cancer	1
		Ovary cancer	1
	Inflammation		
		Takayasu arteritis	2
		Chronic active Epstein–Barr virus infection	1
Time delay (min)			62.2 ± 3.3
Blood sugar level (mg/dL)			119.4 ± 16.1

**Table 2 diagnostics-12-02500-t002:** Qualitative image analysis.

	OSEM	AiCE	*p*-Value
Delineation	3.5 ± 0.63	4.00 ± 0.26	<0.0001 *
Noise	2.77 ± 0.68	3.77 ± 0.68	<0.0001 *
Overall image quality	3.07 ± 0.58	3.83 ± 0.53	<0.0001 *

Data are shown as the mean and standard deviation. OSEM, standard semiconductor PET/CT scanner images; AiCE, semiconductor PET/CT scanner images reconstructed with deep learning. * *p* < 0.05.

**Table 3 diagnostics-12-02500-t003:** SUVs in healthy organ tissues.

Organs		OSEM Mean ± SD	AiCE Mean ± SD	*p*-Value
Parotid gland	SUVmax	1.87 ± 0.58	1.88 ± 0.62	0.281
	SUVpeak	1.62 ± 0.53	1.63 ± 0.53	0.355
	SUVmean	1.33 ± 0.49	1.32 ± 0.50	0.153
Lung	SUVmax	0.57 ± 0.21	0.55 ± 0.21	0.001 *
	SUVpeak	0.49 ± 0.20	0.48 ± 0.20	0.048 *
	SUVmean	0.386 ± 0.152	0.390 ± 0.154	0.009 *
Aortic arch	SUVmax	2.51 ± 0.52	2.50 ± 0.55	0.805
	SUVpeak	2.22 ± 0.54	2.25 ± 0.75	0.555
	SUVmean	1.89 ± 0.43	1.89 ± 0.43	0.807
Left ventricle	SUVmax	3.01 ± 1.08	3.13 ± 1.33	0.159
	SUVpeak	2.95 ± 1.03	3.02 ± 1.21	0.278
	SUVmean	1.94 ± 0.48	1.93 ± 0.49	0.173
Liver	SUVmax	3.61 ± 0.99	3.27 ± 1.03	<0.0001 *
	SUVpeak	3.04 ± 0.86	2.91 ± 0.85	<0.0001 *
	SUVmean	2.56 ± 0.69	2.54 ± 0.68	<0.0001 *
Spleen	SUVmax	2.73 ± 0.61	2.59 ± 0.57	<0.0001 *
	SUVpeak	2.36 ± 0.50	2.31 ± 0.50	<0.0001 *
	SUVmean	2.14 ± 0.47	2.12 ± 0.48	0.001 *
Quadriceps muscle	SUVmax	1.28 ± 0.41	1.04 ± 0.25	<0.0001 *
	SUVpeak	0.93 ± 0.20	0.87 ± 0.19	<0.0001 *
	SUVmean	0.72 ± 0.17	0.72 ± 0.16	0.621

Data are shown as the mean and standard deviation. Note: The current deep convolutional neural network is trained for general whole-body studies but not for the brain. OSEM, standard semiconductor PET/CT scanner images; AiCE, semiconductor PET/CT scanner images reconstructed with deep learning; SD, standard deviation; SUV, standardized uptake value. * *p* < 0.05. SUVmax is the highest SUV, while SUVpeak is the maximum average SUV within a 1-cm^3^ sphere.

**Table 4 diagnostics-12-02500-t004:** SUVs in pathological lesions (*n* = 80).

		OSEM Mean ± SD	AiCE Mean ± SD	*p*-Value
Lesions	SUV max	7.56 ± 5.45	8.99 ± 6.43	<0.0001 *
	SUV peak	4.85 ± 3.04	5.05 ± 3.16	<0.0001 *
	SUV mean	2.17 ± 0.86	2.20 ± 0.91	0.0514

Data are shown as the mean and standard deviation. OSEM, standard semiconductor PET/CT scanner images; AiCE, semiconductor PET/CT scanner images reconstructed with deep learning; SD, standard deviation; SUV, standardized uptake value. * *p* < 0.05. SUVmax is the highest SUV, while SUVpeak is the maximum average SUV within a 1-cm^3^ sphere.

**Table 5 diagnostics-12-02500-t005:** Comparison of lesion detection capability.

		OSEM			
		No Lesion	With Lesions	With More Lesions than AiCE (DLR)	Total
**AiCE (DLR)**	No lesion	0	0	0	0
	With lesions	1	23	0	24
	With more lesions than OSEM	0	6	0	6
	Total	1	29	0	30

DLR, deep learning reconstruction.

## Data Availability

The datasets used and/or analyzed during the current study are available from the corresponding author on reasonable request.

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
