# Peer review of "Enhancement of 18F-Fluorodeoxyglucose PET Image Quality by Deep-Learning-Based Image Reconstruction Using Advanced Intelligent Clear-IQ Engine in Semiconductor-Based PET/CT Scanners"

_diagnostics, 2022, doi:10.3390/diagnostics12102500_

Round 1

Reviewer 1 Report

In this paper, the authors compared the quality of standard semiconductor-based PET/CT scanner images, and the semiconductor PET/CT images reconstructed with deep learning. They scored the comparison from delineation, image noise and overall image quality. Moreover, the qualitative results showed that involved the deep learning reconstruction is better than standard method, which indicates the deep learning technology may be helpful for clinical diagnosis.

This paper could be partially accepted. From the clinical point of view, this paper kept some interesting figures, which indicate the deep learning reconstructed images represent the better performance than standard PET/CT image with much clearer lesions parts. However, the entire article had no description of the architecture of the deep learning model they used. Moreover, the authors did not provide the mathematical description of their quantification methods they used for analyzing results. Hence, the major change should be adding more materials on the deep learning model and formulas of quantification methods to make it easy to understand the meaning of evaluation indicators shown in the paper.

My detailed comments as below.

P.1 Title

The title is Deep leaning-based image reconstruction can enhance ...”, but as far as we know, tens of thousands of deep learning models have been investigated for different targets. In terms of image reconstruction, various models will generate different results with distinct inputs. The results shown in this paper are based on the model of Advanced Intelligent Clear-IQ Engine, so from this perspective, it cannot be generalized to say the deep leaning can enhance the image performance of PET/CT.

P.1 Lines 20-23

Of 21 SUVs in healthy organs ...” This part only has two subjects, but they appear four times among the four sentences, which is repetitive.

P.2 Lines 45-47

It said AiCE, from Canon Medical Systems, is the first commercialized deep learning reconstruction tool for PET/CT imaging. There should be a reference.

P.2 Lines 76-78

Does that mean extending the scan time will get the similarly high-quality images as yield from AiCE?

P.3 Table 1

In Disease” part, what does the integer value on the right-side stand for? If that means the number of patients, does that mean some patients have more than one disease?

P.4 Table 3
What is the difference between SUVmax and SUVpeak? Please make the clarification in the paper. P.5 Table 4
The same question as the last one.

Author Response

Ken Yamagiwa

Reply for reviewer 1

Thank you for taking the time to review our manuscript.

“Hence, the major change should be adding more materials on the deep learning model and formulas of quantification methods to make it easy to understand the meaning of evaluation indicators shown in the paper.”

We agree to add a description of the architecture of the deep learning model. Please see lines 86-91 and Figure 1. About the quantification methods, we added the formula of standardized uptake value (SUV) and the definition of SUVmax, SUVmean, and SUVpeak and provided a reference for the definition of SUVpeak in lines 114-124.

P.1 Title

“The results shown in this paper are based on the model of “Advanced Intelligent Clear-IQ Engine”, so from this perspective, it cannot be generalized to say the deep leaning can enhance the image performance of PET/CT.”

As you pointed out, we don’t think it is possible to generalize that deep leaning can enhance the image quality of semiconductor-based PET/CT. Therefore, we made clear that this study is based on Advanced Intelligent Clear-IQ Engine (AiCE) in the title and replaced the word “DL” by “AiCE” and the word "non-DL" by "OSEM" in the main text.

P.1 Lines 20-23

““Of 21 SUVs in healthy organs ...” This part only has two subjects, but they appear four times among the four sentences, which is repetitive.”

We agree with it and fixed them briefly.

P.2 Lines 45-47

“It said “AiCE”, from Canon Medical Systems, is the first commercialized deep learning reconstruction tool for PET/CT imaging. There should be a reference.”

Since we couldn’t find a precise reference for that, we removed the word “first” from the sentence.

P.2 Lines 76-78

“Does that mean extending the scan time will get the similarly high-quality images as yield from AiCE?”

 Yes, AiCE is aimed to reconstruct high-quality images equivalent to very long scan duration images.

P.3 Table 1

“In “Disease” part, what does the integer value on the right-side stand for? If that means the number of patients, does that mean some patients have more than one disease?”

That means the number of patients. 3 patients had two pathologies. That’s why the total number of diseases was 33. We added this sentence in lines 141-142.

P.4 Table 3
“What is the difference between SUVmax and SUVpeak? Please make the clarification in the paper. P.5”

SUVmax is the SUV of the hottest voxel, while SUVpeak is the maximum average of SUV within a 1-cm3 sphere. We added this sentence in the table caption.

Table 4
“The same question as the last one.”

We added the same sentence in the table caption.

Thanks to your review, we believe our manuscript has become easier for readers to understand.

Reviewer 2 Report

This study aimed to compare the quality of semiconductor-based PET/CT scanner images with and without deep learning based reconstructions. This is a well designed and well written study with very nice results.

Minor comments:

1. The term "lesions" is used through the manuscript to define the pathological foci on PET images. Probably "tumor lesions" or "pathological lesions" will be more clear to the reader.

2. Table 1: there were 30 patients in this study but 33 are mentioned in the table. What were this 3 patients with infection/inflammation status in this cohort? What lesion was seen in the EBV patient?

3. Figure 1: This is a case of a patient with multiple bone metastases. The explanation to the figure is lacking of description of the findings.  What do we see on each reconstruction? Is it better? are there differences between the 2 image types?

4. Figure 2: This figure should show lesions detected on DLR and missed on the non-DLR images. In its current form all lesions are demonstrated on both image sets. Which lesion was missed?

Round 2

Reviewer 1 Report

All the previous comments are addressed, and the details of the deep learning architecture are now included. I have no additional comments.